# Highly Carbon-Resistant Y Doped NiO–ZrO$_m$ Catalysts for Dry Reforming of Methane

Ye Wang [1,2,3], Li Li [2], Yannan Wang [2], Patrick Da Costa [1,*] and Changwei Hu [2,3,*]

[1] Institut Jean Le Rond d'Alembert, Sorbonne Université, CNRS, 2 Place de la Gare de Ceinture, 78210 Saint-Cyr-L'Ecole, France; ye.wang@dalembert.upmc.fr

[2] Key Laboratory of Green Chemistry and Technology, Ministry of Education, College of Chemistry, Sichuan University, Chengdu 610064, China; lili2209362583@163.com (L.L.); yannanwang@mails.ccnu.edu.cn (Y.W.)

[3] College of Chemical Engineering, Sichuan University, Chengdu 610065, China

* Correspondence: patrick.da_costa@upmc.fr (P.D.C.); changweihu@scu.edu.cn (C.H.); Tel.: +33-1-44-27-95-65 (P.D.C. & C.H.)

**Abstract:** Yttrium-doped NiO–ZrO$_m$ catalyst was found to be novel for carbon resistance in the $CO_2$ reforming of methane. Yttrium-free and -doped NiO–ZrO$_m$ catalysts were prepared by a one-step urea hydrolysis method and characterized by Brunauer-Emmett-Teller (BET), TPR-H$_2$, $CO_2$-TPD, XRD, TEM and XPS. Yttrium-doped NiO–ZrO$_m$ catalyst resulted in higher interaction between Ni and ZrO$_m$, higher distribution of weak and medium basic sites, and smaller Ni crystallite size, as compared to the Y-free NiO–ZrO$_m$ catalyst after reaction. The DRM catalytic tests were conducted at 700 °C for 8 h, leading to a significant decrease of activity and selectivity for the yttrium-doped NiO–ZrO$_m$ catalyst. The carbon deposition after the DRM reaction on yttrium-doped NiO–ZrO$_m$ catalyst was lower than on yttrium-free NiO–ZrO$_m$ catalyst, which indicated that yttrium could promote the inhibition of carbon deposition during the DRM process.

**Keywords:** dry reforming of methane; carbon-resistant; yttrium; weak and medium basic sites

## 1. Introduction

Fischer-Tropsch (F-T) synthesis has become a significant process for producing liquid organic hydrocarbons from syngas (H$_2$ and CO). There are several methods to produce syngas: e.g., steam reforming of methane (Equation (1)), partial oxidation of methane (Equation (2)) and dry reforming of methane (Equation (3)) [1–5]. Among these methods, the dry reforming of methane has a competitive advantage of producing clean hydrogen and carbon monoxide mixture gases with an equimolar ratio (1:1), which best suits for F-T synthesis. Moreover, another significant aspect is the consumption of two greenhouse gases ($CO_2$ and $CH_4$), thereby offering an environmental benefit [3,6].

$$CH_4 + H_2O \rightarrow 3H_2 + CO \tag{1}$$

$$2CH_4 + O_2 \rightarrow 2CO + 4H_2 \tag{2}$$

$$CH_4 + CO_2 \rightarrow 2CO + 2H_2 \tag{3}$$

In general, noble metal catalysts (Pt, Ir, Rh) exhibit good performance for the dry reforming of methane [7–9]. Considering the high cost of noble metals for the industrial scale, many efforts have focused on the Ni-based catalysts, because nickel metal has a high potential for industrial application in DRM [10–13]. However, it was well known that Ni-based catalysts suffered deactivation caused by carbon deposition and/or sintering (frittage). Therefore, the development of nickel-based catalysts, with

the significantly promoted Ni-sintering resistance and coke-tolerance, has been extensively investigated in recent years [14–16]. According to the literature [17–19], larger Ni particles on the surface of the catalyst led to catalytic selectivity towards methane cracking (Equation (4)), thereby causing carbon deposition. Another approach of the coke deposition was the disproportionation of CO (Equation (5)).

$$CH_4 \rightarrow 2H_2 + C \qquad\qquad (4)$$

$$2CO \rightarrow CO_2 + C \qquad\qquad (5)$$

It is found that the adjustment of the catalyst by promoters is an efficient approach to restrain the formation of carbon [20–22]. Thus, the modification of the raw catalyst with rare earth elements has been reported to enhance both the activity of the catalyst and inhibition of carbon formation, such as La and Y. The latter has attracted huge interest for reforming catalyst [18,22–24]. Taherian et al. [20] found that comparing with mesoporous Ni/SBA-15, Ni/SBA-15 modified by $Y_2O_3$ can promote the control of the nickel particle size, and thus enhance the dispersion of nickel. Moreover, the Ni-Y/KIT-6 catalyst exhibited better dispersion of nanosized Ni particles inside the pores of the support, as compared to Ni/KIT-6 material, leading to lower carbon deposition in the initial stability test at 700 °C [19]. Li et al. [21] investigated the effect of different types of impregnation for yttrium introduced on the performance of Ni/Y/$Al_2O_3$ catalyst, and observed that compared with the Y-free catalyst, the activity and stability of the Y-modified catalyst increased, because of smaller nickel metallic particle size, more basic sites, the less amount of carbon deposit and the small degree of graphitization. Furthermore, Li et al. [25] also observed high carbon-resistance for the Ni-SBA-15 catalyst modified by yttrium promoter, because of the high dispersion of the nanoparticles of yttrium and nickel. Moreover, more oxygen vacancies were created by the introducing of yttrium.

Except for the active metal and promoter, the support also plays an important role in the catalytic test for DRM. The $ZrO_2$ is considered to be a promising support for the dry reforming of methane, because of the special property, e.g., the reducing and oxidizing abilities, the unexceptionable thermal and chemical stability, and the high oxygen mobilization [22,26–28]. These properties of $ZrO_2$ can be improved by adding a cation with valence lower than 4+, e.g., $Y^{3+}$ [22,23], which results in the formation of solid solution, together with oxygen vacancies. Furthermore, those oxygen vacancies play a key role in gasifying carbon deposits during the activation of $CO_2$ and $O_2$ [23]. Bellido et al. [23] prepared a series of $Y_2O_3$–$ZrO_2$ catalysts with different yttrium content (4 to 12 mol %). 5Ni8YZ (mole fraction of Y = 8 mol %) catalyst showed the best activity and stability for DRM. Because $Y^{3+}$ increased the formation of oxygen vacancies in YZ support. Those modification properties of $ZrO_2$ could improve the performance of catalyst. Asencios et al. [4] investigated the effect of solid solution on performance of NiO–$Y_2O_3$–$ZrO_2$ catalyst and found that the formation of NiO–$Y_2O_3$ and $Y_2O_3$–$ZrO_2$ solid solution could enhance the activity of catalyst for oxidative reforming of model biogas. Moreover, the formation of oxygen vacancies could promote the removal of carbon deposition. Świrk et al. [29,30] also found that the formation of $Y_2O_3$–$ZrO_2$ solid solution could improve the stability of catalyst modified by yttrium.

In order to understand the ability of carbon-resistance on the Y-modified $ZrO_2$ catalyst, the NiO–$ZrO_m$ and NiO–$ZrO_m$–$YO_n$ catalysts were prepared by the one step urea hydrolysis method, and the performance of which was discussed here in detail. This latter catalyst, which was not the most active catalyst in DRM, exhibited, however, a high resistance against carbon deposition. We investigate the ability of carbon resistance on NiO–$ZrO_m$–$YO_n$ catalyst, further to understand the relationship between structure and carbon-resistance on Y-modified catalyst.

## 2. Results and Discussion

### 2.1. The BET Results of Y-Doped and Y-Free NiO–ZrO$_m$ Catalysts

The physical properties of Y-doped and Y-free NiO–ZrO$_m$ material was investigated by the N$_2$ adsorption–desorption method (Table 1). Both of the samples display the same narrow pore size of about 2 nm. The NiO–ZrO$_m$ catalyst shows a surface area of 113 m$^2$/g and a pore volume of 2 nm; while, the NiO–ZrO$_m$–YO$_n$ catalyst exhibits the lower specific surface area (79 m$^2$/g ) and the smaller pore volume (0.1 cm$^3$/g), which may decrease the activity of the catalyst, because the high surface area can enhance the activity of the catalyst [31].

**Table 1.** The results of the Brunauer–Emmett–Teller (BET) experiment for NiO–ZrO$_m$ and NiO–ZrO$_m$–YO$_n$-calcined catalysts, the specific surface area (S$_{BET}$) was determined by the BET method; the pore volume (V$_P$) and the pore diameter (D$_P$) determined by the Barrett–Joyner–Halenda (BJH) method. H$_2$ consumption of calcined catalysts determined by hydrogen temperature programmed reduction (H$_2$-TPR). The content of Zr$^{4+}$ and Zr$^{3+}$ on both catalysts after reduction determined by X-ray photoelectron spectroscopy (XPS), and the content of Ni on both catalysts determined by Inductively Coupled Plasma (ICP) Spectroscopy.

| Catalyst | D$_P$, nm | S$_{BET}$, m$^2$/g | V$_P$, cm$^3$/g | H$_2$ Consumption, mmol H$_2$/g | | | | Zr 3d (%) | | Ni (wt %) |
| --- | --- | --- | --- | --- | --- | --- | --- | --- | --- | --- |
| | | | | Total | Theory [a] | α | β | Zr$^{4+}$ | Zr$^{3+}$ | |
| NiO–ZrO$_m$–YO$_n$ | 2 | 79 | 0.1 | 0.37 | 0.39 | 0.10 | 0.27 | 16 | 84 | 10 |
| NiO–ZrO$_m$ | 2 | 113 | 0.2 | 0.62 | 0.47 | 0.13 | 0.49 | 10 | 90 | 12 |

[a] Caculated by ICP and the comsuption of reduction of pure NiO.

### 2.2. The Reducibility of Y-Doped and Y-Free NiO–ZrO$_m$ Catalysts

The H$_2$-TPR profile of Y-doped and Y-free NiO–ZrO$_m$ catalysts is presented in Figure 1A. Two reduction peaks at about 450 and 650 °C, denominated α and β, respectively, are observed on both catalysts. The first peak (α) corresponds to the reduction of the NiO species of weak and strong interaction with Y and/or Zr [23,27]. This first peak shifts to low temperature on NiO–ZrO$_m$–YO$_n$ catalyst, as compared to the NiO–ZrO$_m$ catalyst. This shift may be attributed to the increase of oxygen vacancies by the addition of the yttrium promoter, because oxygen vacancies can promote the reduction of NiO by weakening the Ni–O bond [32,33]. The β peak on the NiO–ZrO$_m$ catalyst is related to the reduction of the solid solution of NiO–ZrO$_2$ and/or ZrO$_2$, while on the NiO–ZrO$_m$–YO$_n$ catalyst, the β peak may be assigned to the reduction of solid solution (NiO–ZrO$_2$ and/or NiO–Y$_2$O$_3$), ZrO$_2$ and/or surface-capping oxygen ions of the Y$_2$O$_3$–ZrO$_2$ solid solution [4,34]. On the contrary, the β peak shifts to a higher temperature for NiO–ZrO$_m$-YO$_n$ catalyst. Similar phenomenon about the shift was found by Asencios et al. [4] on NiO–Y$_2$O$_3$–ZrO$_2$ catalysts. The yttrium addition could promote the reduction of surface-capping oxygen ions of the NiO–ZrO$_m$–YO$_n$ catalyst, leading to the formation of new surface oxygen vacancies at about 700 °C [4]. Therefore, the β peak shifts to about 700 °C. Except for the shift of peak, the total amount of H$_2$ consumption on the NiO–ZrO$_m$ catalyst (0.62 mmol H$_2$/g) is higher than that on NiO–ZrO$_m$–YO$_n$ catalyst (0.37 mmol H$_2$/g) presented in Table 1. The latter result is consistent with the theoretical value of 0.39 mmol H$_2$/g, while the H$_2$ consumption on this NiO–ZrO$_m$ catalyst is higher than the theoretical value of 0.47 mmol H$_2$/g. This phenomenon proves that the ZrO$_2$ is also reduced by hydrogen. Besides, for the α peak, the amount of H$_2$ consumption on our NiO–ZrO$_m$–YO$_n$ catalyst decreases by adding the yttrium, which indicates that part of this free NiO would be inserted into the structure of ZrO$_2$ and/or Y$_2$O$_3$ to form NiO–ZrO$_2$ and/or NiO–Y$_2$O$_3$ solid solution. Thus, yttrium can promote nickel embedding into the structure of ZrO$_2$ and/or Y$_2$O$_3$. For the β peak, the amount of H$_2$ consumption on the NiO–ZrO$_m$ catalyst is higher than that obtained on NiO–ZrO$_m$–YO$_n$ catalyst. In order to understand the reduction of ZrO$_2$, the Zr 3d peak is resolved into two peaks, which is shown in Figure 1B. The peaks at about 181.4 and 182.3 eV are ascribed to Zr$^{3+}$ and Zr$^{4+}$, respectively [35,36].

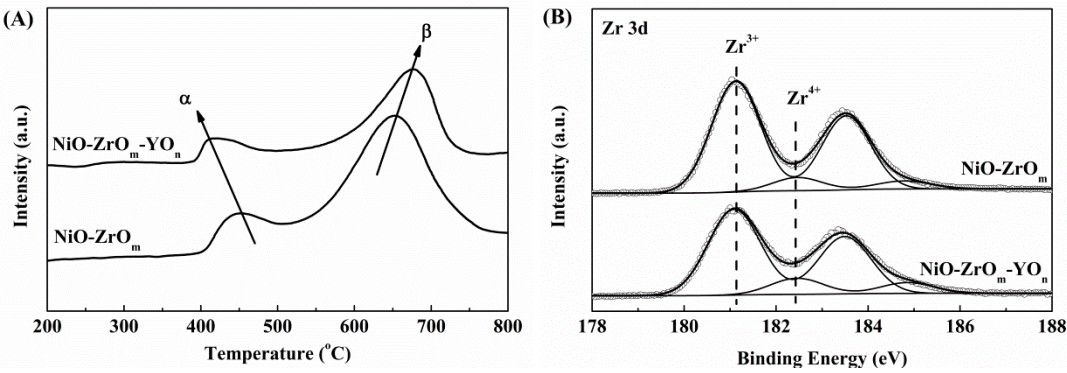

**Figure 1.** (**A**) the $H_2$-TPR profiles of NiO–ZrO$_m$–YO$_n$ and NiO–ZrO$_m$ calcined catalysts, and (**B**) the Zr 3d profiles from XPS measurements of NiO–ZrO$_m$–YO$_n$ and NiO–ZrO$_m$ catalysts after reduction.

It is very obvious to be observed that most $Zr^{3+}$ formed on both catalysts, indicating that the most $Zr^{4+}$ is reduced to $Zr^{3+}$ after reduction. From Table 1, the content of $Zr^{3+}$ on the NiO–ZrO$_m$ catalyst (90%) is higher than that on the NiO–ZrO$_m$–YO$_n$ catalyst (84%), which manifests that more $Zr^{4+}$ is reduced to $Zr^{3+}$ on the NiO–ZrO$_m$ catalyst, which is conformed to $H_2$-TPR results (Figure 1A and Table 1), that is, this NiO–ZrO$_m$ catalyst consumes more $H_2$ during the reduction. These phenomena indicate that the formation of $ZrO_2$ defected by $Ni^{2+}$ on the NiO–ZrO$_m$ catalyst is very easy to be reduced from $Zr^{4+}$ to $Zr^{3+}$ in the presence of hydrogen. While the $ZrO_2$ defected by $Y^{3+}$ is very stable, which is very hard to be reduced. Therefore, the introduction of the $Y^{3+}$ into $ZrO_2$ lattice can also stabilize the crystal structure, and create new oxygen vacancies.

### 2.3. Basicity of Y-Doped and Y-Free NiO–ZrO$_m$ Catalysts

A $CO_2$ temperature-programmed desorption ($CO_2$ TPD) experiment was conducted to determine the basicity of the Y-doped and Y-free NiO–ZrO$_m$ catalysts (Figure 2). The peaks on the NiO–ZrO$_m$–YO$_n$ catalyst shift to low temperature, as compared to the NiO–ZrO$_m$ catalyst. The total number of basic sites increases from 73 to 100 μmol $CO_2$/g on the NiO–ZrO$_m$ catalyst before and after the introducing of yttrium (see in Table 2), which indicates that the yttrium can enhance the total number of basic sites. As already described elsewhere [37–39], there are three types of basic sites (weak, medium-strength and strong). The content of weak peak on the NiO–ZrO$_m$–YO$_n$ catalyst (36.9%) is higher than that on the NiO–ZrO$_m$ catalyst (15.5%), while the content of the strong peak decreases to 13.0% on the NiO–ZrO$_m$–YO$_n$ catalyst. Besides, the position of weak and medium-strength peaks on this NiO–ZrO$_m$–YO$_n$ catalyst are at about 150 and 240 °C, which are lower than those on the NiO–ZrO$_m$ catalyst (185 and 252 °C), respectively. This phenomenon manifests that yttrium can promote the formation of weak basic sites on the NiO–ZrO$_m$–YO$_n$ catalyst. Thus, one can note that the addition of yttrium modifies both the distribution and the number of basic sites. A similar phenomenon can be observed in our group's previous works [37,40]. According to the literature [10,38,39], the weak and medium-strength basic sites can promote the formation of activation carbonate species, thereby enhancing the ability to remove the carbon deposition. Whereas, too strong basic sites lead to too strong $CO_2$ adsorption, thereby promoting more carbon deposition. Therefore, the weak and medium basic sites present on the NiO–ZrO$_m$–YO$_n$ catalyst can enhance the ability to eliminate coke.

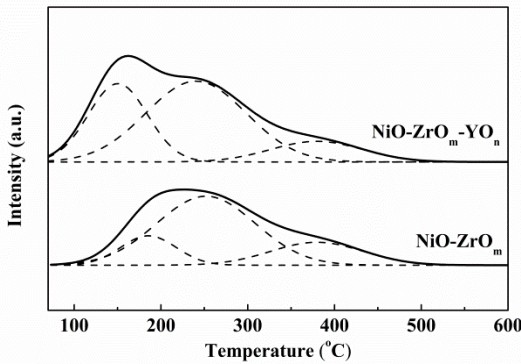

**Figure 2.** The $CO_2$-TPD profiles of the NiO–$ZrO_m$–$YO_n$ and NiO–$ZrO_m$ catalysts.

**Table 2.** Basicity measured by the content of $CO_2$ desorption on $CO_2$-temperature programmed desorption experiment over the NiO–$ZrO_m$–$YO_n$ and NiO–$ZrO_m$ catalysts.

| Catalyst | CO$_2$ Peak Identification | | | | | | Total Basicity (μmol CO$_2$/g) |
|---|---|---|---|---|---|---|---|
| | Peak 1 (Weak) | | Peak 2 (Medium-Strength) | | Peak 3 (Strong) | | |
| | Position (°C) | Content (%) | Position (°C) | Content (%) | Position (°C) | Content (%) | |
| NiO–$ZrO_m$ | 185 | 15.5 | 252 | 64.4 | 380 | 20.1 | 73 |
| NiO–$ZrO_m$–$YO_n$ | 150 | 36.9 | 240 | 55.4 | 380 | 13.0 | 100 |

*2.4. Nickel Particle Size and Crystallized Phases of Y-Doped and Y-Free NiO–$ZrO_m$ Catalysts*

Figure 3 shows X-ray diffraction (XRD) patterns of the NiO–$ZrO_m$–$YO_n$ and NiO–$ZrO_m$ catalysts after reduction and after catalytic reaction. Tetragonal and/or cubic phase $ZrO_2$ appear in both catalysts. The crystallite sizes of $ZrO_2$ on both catalysts exhibit the same value of 7 nm from the Scherrer Equation, and do not change even after reaction for 8 h (Table 3). The peak at about 44.5° can be attributed to the metallic nickel [41]. The $Ni^0$ size on the NiO–$ZrO_m$ catalyst is about 12 nm after reduction, and it increases to 24 nm after reaction. While for the NiO–$ZrO_m$–$YO_n$ catalyst, it decreases from 16 nm to 10 nm after reaction for 8 h, which indicates the re-dispersion of $Ni^0$. Similar phenomenon had been reported by other researches [37,38,42]. Nakayam et al. [43] found the re-dispersion of Ni under an alternating condition between $H_2$ reduction and oxidation atmosphere. Under dry reforming of methane condition, $CO_2$ is a source of oxygen. With $CO_2$ adsorption and activation on the surface of catalyst, the nickel metal may be oxidized. The production of $H_2$ and CO as reduction atmosphere may contribute to the reduction of the NiO, and thereby the re-dispersion of nickel particles. On one hand, the $Ni^0$ size is related to the sintering of the nickel. Severe sintering takes place on this NiO–$ZrO_m$ catalyst, due to the lower interaction between Ni and Zr, which is comfirmed by the results of $H_2$-TPR. Furthermore, the addition of yttrium can limit the sintering of nickel during the reaction. On the other hand, It is well known that the large nickel particle size may be favored for selective reactions that lead to carbon deposition [18,19]. As a consequence, NiO–$ZrO_m$–$YO_n$ can limit the carbon deposition. Besides, NiO–$ZrO_m$–$YO_n$ exhibits more basic sites, which could enhance the ability of the adsorption of $CO_2$, thereby promoting the removal of carbon deposition. Both sintering and carbon deposition could contribute to the deactivation of catalyst, thereby reducing the stability of the catalyst.

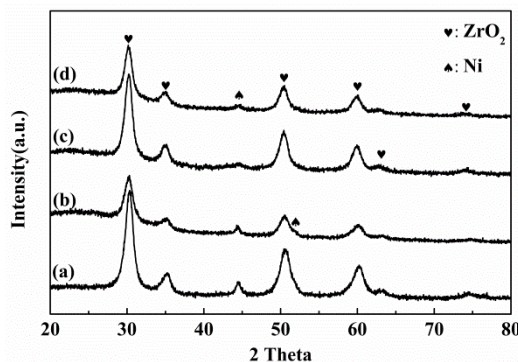

**Figure 3.** The X-ray diffraction (XRD) profiles of (**a**) NiO–ZrO$_m$ and (**c**) NiO–ZrO$_m$–YO$_n$ catalysts after reduction at 700 °C for 1 h and after reaction at 700 °C for 8 h, and (**b**) NiO–ZrO$_m$ and (**d**) NiO–ZrO$_m$–YO$_n$ catalysts after reaction at 700 °C for 8 h.

**Table 3.** Crystallite sizes of ZrO$_2$ and Ni$^0$ on NiO–ZrO$_m$–YO$_n$ and NiO–ZrO$_m$ catalysts after reduction at 700 °C for 1 h and after reaction at 700 °C for 8 h, determined by XRD. The particle size of nickel after reaction for 8 h determined by transmission electron microscopy (TEM). The coke content after reaction for 8 h was determined by thermogravimetric analysis (TGA). The C–C content after reaction for 8 h determined by X-ray photoelectron spectroscopy (XPS). The I$_G$/I$_D$ after reaction for 8 h determined by Raman spectroscopy.

| Catalyst | ZrO$_2$ (nm) | | Ni$^0$ (nm) | | Particle Size (nm) | Coke Content (%) | C–C Content (%) | I$_G$/I$_D$ |
|---|---|---|---|---|---|---|---|---|
| | Reduction | Reaction | Reduction | Reaction | | | | |
| NiO–ZrO$_m$ | 7 | 7 | 12 | 24 | 15–20 | 3.7 | 42 | 1.7 |
| NiO–ZrO$_m$–YO$_n$ | 7 | 7 | 16 | 10 | 10–15 | 1.0 | 38 | - |

## 2.5. The Performance of Y-Doped and Y-Free NiO–ZrO$_m$ Catalysts

The catalytic performance of the catalysts was investigated at 700 °C for 8 h (Figure 4). The conversion of methane on both catalysts decreases within 1 h, and finally stabilizes at 67% and 85% for the NiO–ZrO$_m$–YO$_n$ and NiO–ZrO$_m$ catalyst, respectively. Except for the higher methane conversion on the NiO–ZrO$_m$ catalyst, the CO$_2$ conversion and the ratio of H$_2$/CO are higher than those on the NiO–ZrO$_m$–YO$_n$ catalyst. The CO$_2$ conversion on the NiO–ZrO$_m$ catalyst is about 89% with the H$_2$/CO ratio of 0.95, which is very close to one. When adding yttrium into the NiO–ZrO$_m$ catalyst, the CO$_2$ conversion decreases within 60 min and stabilizes at 70% for 8 h time on stream. At the same time, the H$_2$/CO ratio decreases to about 0.85. Because the NiO–ZrO$_m$–YO$_n$ catalyst exhibits the lower specific surface area, the smaller pore volume, and the bigger metallic nickel particle size. Those properties could lead to the lower performance of this NiO–ZrO$_m$–YO$_n$ catalyst. However, the catalyst activity remained stable during time on stream.

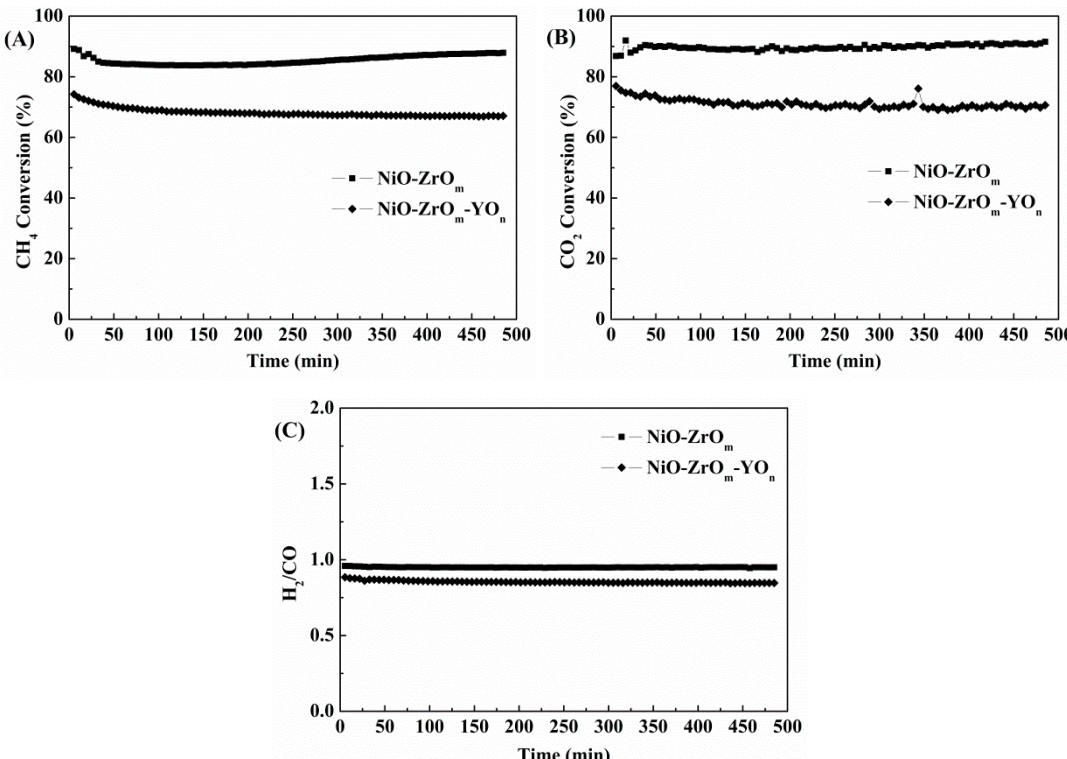

**Figure 4.** The conversion of $CH_4$ (**A**) and $CO_2$ (**B**), and the $H_2/CO$ ratio (**C**) of $NiO–ZrO_m–YO_n$ and $NiO–ZrO_m$ catalysts at 700 °C with the mixed flow of $CH_4:CO_2:Ar = 10:10:80$, GSHV = 48,000 h$^{-1}$.

*2.6. On the Carbon Deposition on Used Y-Doped and Y-Free NiO–ZrO$_m$ Catalysts*

2.6.1. Carbon Formation Evidenced by XPS and Raman Spectroscopy

The content of coke on both catalysts are determined by TGA experiment, and the results are shown in Table 3. The content of the coke on the $NiO–ZrO_m$ catalyst is about 3.7%, which is higher than that on $NiO–ZrO_m–YO_n$ catalyst (1.0%), indicating that yttrium can decrease the carbon deposition on the $NiO–ZrO_m–YO_n$ catalyst. Figure 5A represents the results of C 1s profiles. The intensity of C 1s on the $NiO–ZrO_m–YO_n$ catalyst is lower than that on our $NiO–ZrO_m$ catalyst, showing that the content of surface carbon deposition on this $NiO–ZrO_m–YO_n$ catalyst is lower, about 63%, While the coke on the surface of the $NiO–ZrO_m$ catalyst is about 69%. Figure 5B shows the results of the Raman experiment. Two peaks can be found on the $NiO–ZrO_m$ catalyst, and no peak can be observed on the $NiO–ZrO_m–YO_n$ catalyst, which manifests that no or less coke form on the $NiO–ZrO_m–YO_n$ catalyst. The peak at about 1328 cm$^{-1}$ is attributed to the structural imperfections on not-organized carbon materials, namely disorder-induced band (D band), and another peak at about 1585 cm$^{-1}$ is corresponded to the in-plane C–C stretching vibrations of sp$^2$ atoms in coke, namely graphitic carbon (G band) [16,44,45]. The intensity of peak is named I, while the ratio of $I_G/I_D$ is about 1.7 (Table 3), indicating that more graphitic carbon forms on the $NiO–ZrO_m$ catalyst, which is conformed to the results of XPS, more C–C species on the $NiO–ZrO_m$ catalyst. All those phenomena show that the carbon deposition on this $NiO–ZrO_m$ catalyst is higher than that on the $NiO–ZrO_m–YO_n$ catalyst.

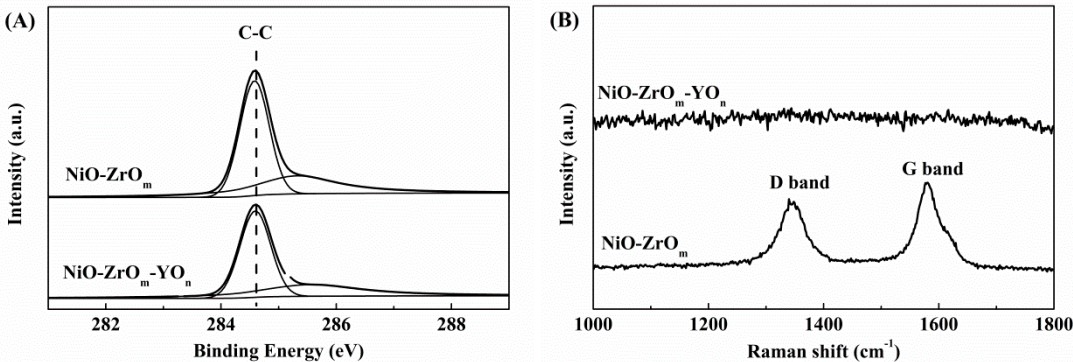

**Figure 5.** The C 1s (**A**) and Raman (**B**) profiles of used NiO–ZrO$_m$ and NiO–ZrO$_m$–YO$_n$ catalysts.

### 2.6.2. On the Study of the Morphology of Carbon Studied by TEM

The morphology structure of carbon on the catalysts after reaction is determined by the transmission electron microscopy experiment (Figure 6). After reaction, the nickel particle size decreases on NiO–ZrO$_m$ by adding the yttrium promoter (Figure 6A,C). The nickel particle size formed on NiO–ZrO$_m$–YO$_n$ is about 10–15 nm (Figure 6D), while about 15–20 nm becomes formed on NiO–ZrO$_m$ (Figure 6B), which is corresponding to the results of XRD. It can be noted that large Ni particles (over 20 nm) can be observed on the NiO–ZrO$_m$ catalyst. According to the results of H$_2$-TPR, the interaction between Ni and ZrO$_m$ on the NiO–ZrO$_m$ catalyst is lower than that on the NiO–ZrO$_m$–YO$_n$ catalyst, leading to nickel sintering. Therefore, bigger particles of Ni$^0$ are present on the NiO–ZrO$_m$ catalyst. These large nickel particles may be favored for selective reactions that tend to form carbon deposition [18,19], thereby resulting in the formation of carbon deposition. Thus, a lot of carbon deposition can be observed on the NiO–ZrO$_m$ catalyst in the form of a carbon nanotube (Figure 6A), and no carbon deposition is observed on the NiO–ZrO$_m$–YO$_n$ catalyst (Figure 6C),which indicates that no or little carbon formes on the NiO–ZrO$_m$–YO$_n$ catalyst. These carbon nanotubes are graphitic carbon, which is in agreement with the results of the Raman experiment. This phenomenon shows that yttrium can suppress the sintering of nickel particles, and also inhibits the formation carbon during the DRM reaction.

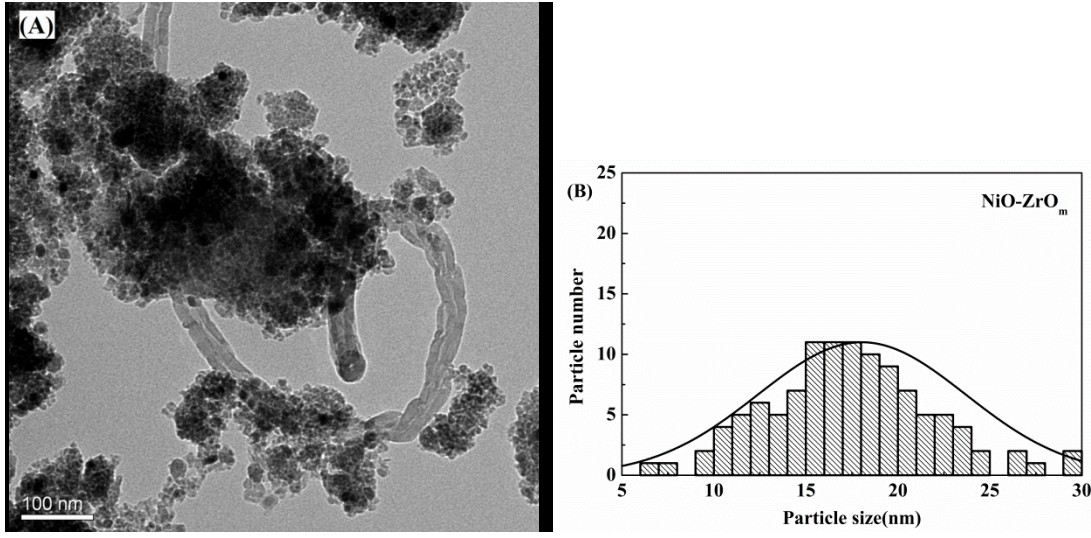

**Figure 6.** *Cont.*

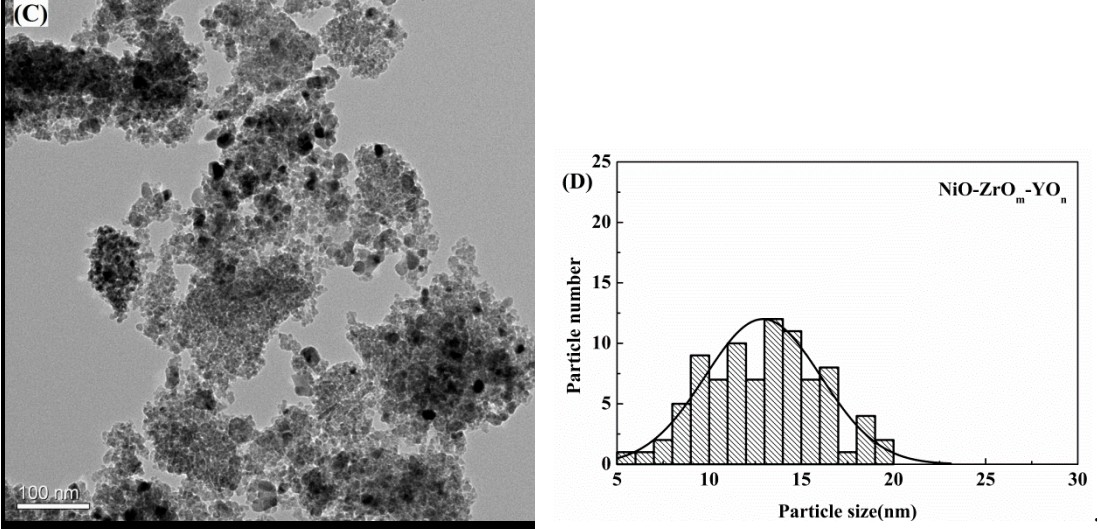

**Figure 6.** Transmission electron microscopy (TEM) images of used catalysts (**A**) NiO–ZrO$_m$ and (**C**) NiO–ZrO$_m$–YO$_n$ and particle sizes of catalysts (**B**) NiO–ZrO$_m$ and (**D**) NiO–ZrO$_m$–YO$_n$.

## 3. Materials and Methods

### 3.1. Synthesis of Y Doped NiO–ZrO$_m$ and NiO–ZrO$_m$ Catalysts

5.03 g zirconium(IV) oxynitrate hydrate (Sigma-Aldrich, Saint-Quentin Fallavier, France), 1.14 g nickel(II) nitrate hexahydrate (Emsure, Merck, Fontenay sous Bois, France) 7.06 g urea (Sigma-Aldrich, Saint-Quentin Fallavier, France), 0.86 g yttrium(III) nitrate hexahydrate (Sigma-Aldrich, Saint-Quentin Fallavier, France) (corresponding to a loading of 10 wt %) and 7.96 g of Pluronic P123 (Sigma-Aldrich, Saint-Quentin Fallavier, France) amphiphilic block copolymer were dissolved in 375 mL of distilled water. The mixed liquor was heated from room temperature to 95 °C with constant stirring for 48 h. After that, the mixture was aged at 100 °C for 24 h. Then the slurry was filtered, washed with a little of distilled water and dried at room temperature. The catalysts were calcined in air at 800 °C for 5 h with an increasing rate of 1 °C/min. The obtained materials were denoted as NiO–ZrO$_m$–YO$_n$ (Y$_{wt\%}$ = 10%). In order to understand the function of yttrium, NiO–ZrO$_m$ without yttrium was prepared by the above method and denoted as NiO–ZrO$_m$.

### 3.2. Activity Test

The activity test was carried out at 700 °C in a fixed-bed flow reactor (id = 12 mm) connected in-line to a gas micro chromatograph (Agilent Varian GC490, Agilent, Les Ulis, France), equipped with a thermal conductivity detector (TCD). The total feed gas flow rate was 100 mL/min with a molar ratio CH$_4$/CO$_2$/Ar = 1/1/8. Considering the volumes of NiO–ZrO$_m$ and NiO–ZrO$_m$–YO$_n$ catalysts, the total gas hourly space velocity (GHSV) values were 48,000 h$^{-1}$. Prior to reactions, the sample was reduced with 5% H$_2$/Ar flow at 700 °C for 1 h. The CO$_2$ and CH$_4$ conversion, and H$_2$/CO molar ratio of catalysts were calculated as follows:

$$X_{CH_4} = \frac{n_{CH_4,in} - n_{CH_4,out}}{n_{CH_4,in}} \times 100\% \tag{6}$$

$$X_{CO_2} = \frac{n_{CO_2,in} - n_{CO_2,out}}{n_{CO_2,in}} \times 100\% \tag{7}$$

$$H_2/CO = n_{H_2,out}/n_{CO,out} \tag{8}$$

where $X_{CH_4}$ and $X_{CO_2}$ refers to the conversion of CH$_4$ and CO$_2$.

*3.3. Catalyst Characterization*

The physical property was obtained by the $N_2$ adsorption–desorption method. Before measurement, the sample was pretreated under vacuum conditions at 200 °C for 2 h. Then the test conducted in a Belsorp Mini II apparatus (BEL Japan) instrument under liquid nitrogen temperature (−196 °C). The surface area was calculated by the Brunauer–Emmett–Teller (BET) method, and the pore volume and diameter were calculated by the Barrett–Joyner–Halenda (BJH) method.

Temperature-programmed reduction of $H_2$ ($H_2$-TPR) conducted on a BELCAT-M (BEL Japan, BEL Europe GmbH, Krefeld, Germany) apparatus, equipped with a TCD. The sample (60 mg) was pretreated under helium atmosphere at 150 °C for 30 min. The sample was reduced under 5% $H_2$/Ar mixture flow. The temperature increased from 100 °C to 900 °C with a heating rate of 10 °C/min and held on 30 min at 900 °C.

Temperature-programmed desorption of $CO_2$ ($CO_2$-TPD) was conducted on the same equipment. The sample was reduced under a 5% $H_2$/Ar flow at 700 °C for 1 h. Afterward, $CO_2$ was adsorbed at 80 °C for 1 h under a mixture of 10% $CO_2$ in He. After cleaning the weakly adsorbed $CO_2$ for 30 min, the sample was heated to 900 °C with a heating rate of 10 °C/min under a He flow. The obtained graphs were fitted into three peaks (weak, middle and strong basic sites).

X-ray diffraction (XRD) experiment was carried out on a DX-1000 CSC diffractometer, equipped with Cu K$\alpha$ radiation source. The data was recorded in a range of $10° < 2\theta < 80°$, with a scan step size of 0.03°.

Transmission electron microscopy (TEM) experiments were measured on an FEI Tecnai G2 20 Twin instrument at an acceleration voltage of 200 kV.

X-ray photoelectron spectroscopy (XPS) experiment conducted on a KRATOS spectrometer with an AXIS Ultra DLD.

Thermogravimetric analysis (TGA) was used to characterize the carbon deposition of used catalysts. The sample (10 mg) was treated under air atmosphere with a flow rate of 30 mL/min$^{-1}$ at room tempeature until the scales balanced. Then, the temperature increased from room temperature to 800 °C with a heating rate of 5 °C/min$^{-1}$. The data was recorded (the weight of sample) from room temperature to 800 °C.

Raman spectroscopy measurements were carried out on an objective (X50LWD) with a Filter of D1, a Hole of 200 μm, a Grating of 600 gr/mm and a Laser of 532.17 nm. The wavenumber values were scaned over the range 40−4000 cm$^{-1}$ for three times.

## 4. Conclusions

NiO–ZrO$_m$ and NiO–ZrO$_m$–YO$_n$ catalysts were prepared by the urea hydrolysis method and were characterized by BET, TPR-$H_2$, $CO_2$-TPD, XRD, TEM and XPS. After adding yttrium, the strong interaction between Ni and ZrO$_m$ formed on the NiO–ZrO$_m$–YO$_n$ catalyst, resulting in the small size of nickel particle distributed on catalyst after reaction. While for the NiO–ZrO$_m$ catalyst, large particles formed on catalyst after reaction, contributing to the deposition of carbon. Besides, a greater amount of weak and medium-strong basic sites formed on the NiO–ZrO$_m$–YO$_n$ catalyst, which could enhance the ability of the removal of carbon. Except for the advantages, the NiO–ZrO$_m$–YO$_n$ catalyst showed lower specific surface area and the smaller pore volume. Thus, from the results of the activity test at 700 °C for 8 h, our NiO–ZrO$_m$ catalyst exhibited higher methane and $CO_2$ conversion, while the NiO–ZrO$_m$–YO$_n$ catalyst exhibited high carbon resistance for dry reforming of methane at 700 °C. Therefore, yttrium can modify the interaction between Ni and ZrO$_m$, in enhancing the dispersion of nickel particles during the reaction, and promote the formation of more weak and medium-strength basic sites.

**Author Contributions:** Y.W. (Ye Wang) did all the tests, the methodology and did the writing of original draft, L.L. participated in the characterization part and participated in the writing of original draft, Y.W. (Yannan Wang) participated in the characterization part and participated in the writing of original draft, P.D.C. participated in the

writing of the original draft and the funding acquisition and the supervision, C.H. participated in the writing of original draft and the funding acquisition and the supervision.

**Funding:** This work was supported by the National Key R&D Program of China (2018YFB1501404), the 111 program (B17030) and Fundamental Research Funds for the Central Universities.

**Acknowledgments:** Ye Wang acknowledges the financial support of CSC (China Scholarship Council) for her joint-PhD research in Sorbonne Université. We thank Yunfei Tian of analytical & testing center of Sichuan University for XPS experiments.

**Conflicts of Interest:** The authors declare no conflict of interest.

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
