# Peer review of "Highly Carbon-Resistant Y Doped NiO–ZrOm Catalysts for Dry Reforming of Methane"

_catalysts, doi:10.3390/catal9121055_

Round 1

Reviewer 1 Report

This work by Wang et al., outlined the use of ZrOand Y-addition for Ni catalyst supports for the dry reforming of methane. Whilst the work shows some insights into the role of Y-doping, there are some flaws and novelty questions which require addressing prior to consideration for publication. 

In the introduction it is mentioned that Bellido studied a series of Y2O3-ZrO2 materials as Ni support for the dry reforming of methane. In this case, what is the benefit and novelty of this work?  It is not valid to compare the H2 consumption, and attribute it to varying support reducibility without providing the actual Ni loading of the samples. If the Ni loading is not equal, much of the TPR discussion in invalid. It is recommended that ICP be done to confirm the loading, and the amount of Y in the sample It is key that the Ni size changes for the Y and non-doped samples. Why does the size decrease after reaction in the case of the Y? Is it merely a Ni size impact that increases the stability or does the CO2 interaction play a role? The XRD discussion is also flawed. The “weakening” of peaks after reaction is only a function of quantity present and the  widening of the peaks after reaction would be picked up by FWHM and thus crystallite size which remains unchanged. Lines 190-191 are very confusing and contradictory. The samples have the same pore size and the Y-containing sample has smaller sizes after reaction For the TEM results in Table 3, the particle size of what and under what conditions?

Other smaller points include:

Some errors in referencing (e.g. Reference [10] in the introduction does not use ZrO2 as a support as suggested by the authors) The English needs fixing throughout. The tenses have issues and require correction. Examples include: Line 42 "Because nickel metal is highly potential for industrial application in DRM "  Line 43 "43 However, it is well known that Ni-based catalysts suffered from catalyst deactivation that is caused …" Line 48 "Another approach of the coke deposition is the disproportionation of CO" Line 88 "The physical properties of Y-doped and Y-free NiO-ZrOm material are investigated by the N2 89 adsorption-desorption method (Table 1)" Table 1 – needs work. What does D and S mean. These are not common acrynomns, D (referring to pore size diameter I have not seen). What do alpha and beta stand for? It is mentioned below but needs to be in the caption of the title. Figure 3, what is each a, b, c and d? The samples are referred to differently throughout (Ni-Zr or NiO-ZrO)

Author Response

Responses to Reviewer 1:

Comments:

Reviewer 1:

This work by Wang et al., outlined the use of ZrO2 and Y-addition for Ni catalyst supports for the dry reforming of methane. Whilst the work shows some insights into the role of Y-doping, there are some flaws and novelty questions which require addressing prior to consideration for publication.

In the introduction it is mentioned that Bellido studied a series of Y2O3-ZrO2 materials as Ni support for the dry reforming of methane. In this case, what is the benefit and novelty of this work?

Answer: Thank you for your nice comments! The work by Bellido et al was a nice investigation of the Y2O3-ZrO2 materials as support. The novelty of this work is that NiO-ZrOm and NiO-ZrOm-YOn were prepared by one-step synthesis method. We investigate here also the ability of carbon resistance on NiO-ZrOm-YOn catalyst, further to understand the relationship between structure and carbon-resistance on Y-modified catalyst. We had revised this part of the manuscript.

It is not valid to compare the H2 consumption, and attribute it to varying support reducibility without providing the actual Ni loading of the samples. If the Ni loading is not equal, much of the TPR discussion in invalid. It is recommended that ICP be done to confirm the loading, and the amount of Y in the sample

Answer: Thank you very much for your kind suggestion. We have added the ICP experiments and have revised the relevant part of the manuscript.

Table 1 The results of the BET experiment for NiO-ZrOm and NiO-ZrOm-YOn calcined catalysts, the specific surface area (SBET) was determined by BET method; the pore volume (VP) and the pore diameter (DP) determined by BJH method. H2 consumption of calcined catalysts determined by H2-TPR. The content of Zr4+ and Zr3+ on both catalysts after reduction determined by XPS. And the content of Ni on both catalysts determined by ICP.

Catalyst

DP,

nm

SBET, m2/g

VP, cm3/g

H2 consumption, mmol H2/g

Zr 3d (%)

Ni

(wt %)

Total

Theorya

α

β

Zr4+

Zr3+

NiO-ZrOm-YOn

2

79

0.1

0.37

0.39

0.10

0.27

16

84

10

NiO-ZrOm

2

113

0.2

0.62

0.47

0.13

0.49

10

90

12

a : Caculated by ICP and the comsuption of reduction of pure NiO

Except for the shift of peak, the total amount of H2 consumption on NiO-ZrOm catalyst (0.62 mmol H2/g) is highter than that on NiO-ZrOm-YOn catalyst (0.37 mmol H2/g). The latter result is consistent with the theoretical value of 0.39 mmol H2/g, while the H2 consumption on NiO-ZrOm catalyst is higher than the theoretical value of 0.47 mmol H2/g. This phenomenon proves that the ZrO2 is also reduced by hydrogen.

It is key that the Ni size changes for the Y and non-doped samples. Why does the size decrease after reaction in the case of the Y? Is it merely a Ni size impact that increases the stability or does the CO2 interaction play a role?

Answer: Thank you very much for your kind suggestion.

(1) The Ni0 size on NiO-ZrOm catalyst is about 12 nm after reduction, it increases to 24 nm after reaction. While for NiO-ZrOm-YOn catalyst, and it decreases from 16 nm to 10 nm after reaction for 8 hours, which indicates the re-dispersion of Ni0. Similar phenomenon had been reported by other researches [1-3]. Nakayam et al. [4] found the re-dispersion of Ni under an alternating condition between H2 reduction and oxidation atmosphere. Under dry reforming of methane condition, CO2 is a source of oxygen. When CO2 adsorption and activation on the surface of catalyst, the nickel metal may be oxidized. The production of H2 and CO as reduction atmosphere, may contribute to the reduction of the NiO, and thereby the re-dispersion of nickel particles.

(2) On one hand, the Ni0 size is related to the sintering of nickel. Severe sintering takes place on NiO-ZrOm catalyst, due to the lower interaction between Ni and Zr, which is comfirmed by the results of H2-TPR. Furthermore, the addition of yttrium can limit the sintering of nickel during the reaction. On the other hand, it is well known that the large nickel particle size maybe favors selective reactions that lead to form carbon deposition [5, 6]. As a consequence, NiO-ZrOm-YOn can limit the carbon deposition. Besides, NiO-ZrOm-YOn exhibits more basic sites, which could enhance the ability of adsorption of CO2, thereby promoting the removal of carbon deposition. both sintering and carbon deposition could contribute to the deactivation of catalyst, thereby reducing the stability of the catalyst.

The XRD discussion is also flawed. The “weakening” of peaks after reaction is only a function of quantity present and the widening of the peaks after reaction would be picked up by FWHM and thus crystallite size which remains unchanged.

Answer: Thank you for pointing out our imprudent presentation! Fig. 3 shows XRD patterns of NiO-ZrOm-YOn and NiO-ZrOm catalysts after reduction and after catalytic reaction. Tetragonal and/or cubic phase ZrO2 appear in both catalysts. The crystallite sizes of ZrO2 on both catalysts exhibits the same value of 7 nm from Scherrer Equation and does not change even after reaction for 8 hours (Table 3). We have modified it in the revised manuscript.

Lines 190-191 are very confusing and contradictory.

Answer: Thank you for pointing out this point! Lines 190-191 have been modified to When adding yttrium into NiO-ZrOm catalyst, the CO2 conversion decreases within 60 min and stabilizes at 70 % for 8 hours time on stream. At the same time, the H2/CO ratio decreases to about 0.85. We had revised the relevant part of the manuscript.

The samples have the same pore size and the Y-containing sample has smaller sizes after reaction For the TEM results in Table 3, the particle size of what and under what conditions?

Answer: Thank you for your nice comments! We counted the partcile size of nickel after reaction for 8 hours. We have modified it in the Table 3.

Table 3 Crystallite sizes of ZrO2 and Ni0 on NiO-ZrOm-YOn and NiO-ZrOm catalysts after reduction at 700 °C for 1 hour and after reaction at 700 °C for 8 hours, determined by XRD. The particle size of nickel after reaction for 8 hours determined by TEM. The coke content after reaction for 8 hours was determined by TGA. The C-C content after reaction for 8 hours determined by XPS. The IG/ID after reaction for 8 hours determined by Raman spectroscopy.

Catalyst

ZrO2 (nm)

Ni0 (nm)

Particle

Coke

Content

(%)

C-C

content (%)

IG/ID

Reduction

Reaction

Reduction

Reaction

Size (nm)

NiO-ZrOm

7

7

12

24

15-20

3.7

42

1.7

NiO-ZrOm-YOn

7

7

16

10

10-15

1.0

38

-

Other smaller points include:

Some errors in referencing (e.g. Reference [10] in the introduction does not use ZrO2 as a support as suggested by the authors) The English needs fixing throughout. The tenses have issues and require correction. Examples include: Line 42 "Because nickel metal is highly potential for industrial application in DRM " Line 43 "43 However, it is well known that Ni-based catalysts suffered from catalyst deactivation that is caused …" Line 48 "Another approach of the coke deposition is the disproportionation of CO" Line 88 "The physical properties of Y-doped and Y-free NiO-ZrOm material are investigated by the N2 89 adsorption-desorption method (Table 1)" Table 1 – needs work. What does D and S mean. These are not common acrynomns, D (referring to pore size diameter I have not seen). What do alpha and beta stand for? It is mentioned below but needs to be in the caption of the title. Figure 3, what is each a, b, c and d? The samples are referred to differently throughout (Ni-Zr or NiO-ZrOm)

Answer: Thanks for pointing out our error!

(1) We had deleted the Ref. 10 from the related part and had checked all the Reference. We had corrected the English in the text.

(2) We have modified the sentence in line 42 to Considering the high cost for the industrial scale, many efforts have focused on the Ni-based catalysts; because nickel metal has a high potential for industrial application in DRM.

(3) Line 43 has been revised to However, it was well known that Ni-based catalysts suffered deactivation caused by carbon deposition and/or sintering.

(4) Line 48 has been revised to Another approach of the coke deposition was the disproportionation of CO (Equation (5)).

(5) Line 88 and 89 has been revised to The physical properties of Y-doped and Y-free NiO-ZrOm material was investigated by the N2 adsorption-desorption method (Table 1).

(6) Table 1 has been revised as Table 1 The results of the BET experiment for NiO-ZrOm and NiO-ZrOm-YOn calcined catalysts, the specific surface area (SBET) was determined by BET method; the pore volume (VP) and the pore diameter (DP) determined by BJH method. H2 consumption of calcined catalysts determined by H2-TPR. The content of Zr4+ and Zr3+ on both catalysts after reduction determined by XPS. And the content of Ni on both catalysts determined by ICP.

Catalyst

DP,

nm

SBET, m2/g

VP, cm3/g

H2 consumption, mmol H2/g

Zr 3d (%)

Ni

(wt %)

Total

Theorya

α

β

Zr4+

Zr3+

NiO-ZrOm-YOn

2

79

0.1

0.37

0.39

0.10

0.27

16

84

10

NiO-ZrOm

2

113

0.2

0.62

0.47

0.13

0.49

10

90

12

a : Caculated by ICP and the comsuption of reduction of pure NiO

(7) The title of Figure 3 has been revised as Figure 3 The XRD profiles of (a) NiO-ZrOm and (c) NiO-ZrOm-YOn catalysts after reduction at 700 °C for 1 hour and after reaction at 700 °C for 8 hours, and (b) NiO-ZrOm and (d) NiO-ZrOm-YOn catalysts after reaction at 700 °C for 8 hours.

Reference:

[1] R. Dębek, M. Motak, M.E. Galvez, T. Grzybek, P. Da Costa, Influence of Ce/Zr molar ratio on catalytic performance of hydrotalcite-derived catalysts at low temperature CO2 methane reforming, Int. J. Hydrogen Energy, (2017) 42, 1–12

[2] Świrk K., Galvez M.E., Motak M., Grzybek T., Rønning M., Costa P. Da, Yttrium promoted Ni-based double-layered hydroxides for dry methane reforming, J. CO2 Util. (2018) 27, 247-258

[3] Świrk K, Rønning M, Motak M, et al. Ce-and Y-Modified Double-Layered Hydroxides as Catalysts for Dry Reforming of Methane: On the Effect of Yttrium Promotion. Catalysts (2019) 9, 56.

[4] D. Li, K. Nishida, Y. Zhan, T. Shishido, Y. Oumi, T. Sano, K. Takehira, Sustainable Ru-doped Ni catalyst derived from hydrotalcite in propane reforming, Appl. Clay Sci. (2009) 43, 49–56.

[5] T. Nakayama, M. Arai, Y. Nishiyama, Dispersion of nickel particles supported on alumina and silica in oxygen and hydrogen, J. Catal. (1984) 87, 108–115.

[6] Świrk K., Gálvez M.E., Motak M., Grzybek T., Rønning M., Costa P. Da, Syngas production from dry methane reforming over yttrium-promoted nickel-KIT-6 catalysts, Int. J. Hydrogen. Energy, 44 (2019) 274-286

Reviewer 2 Report

This article is very well written, concise, and has its conclusions firmly justified by the experiments. 

Before publishing, please add to the experimental the details of the TG-MS and the Raman experiments since these were used to quantify coking. 

I also believe that the quality of the manuscript would be improved if CO adsorption tests were done as well. This would give the authors information about the Ni dispersion as a function of Y addition. This group seems to have all of the necessary equipment to perform such an experiment based on the CO2/N2 adsorption results already presented in the paper. 

Author Response

Reviewer 2:

This article is very well written, concise, and has its conclusions firmly justified by the experiments. Before publishing, please add to the experimental the details of the TG-MS and the Raman experiments since these were used to quantify coking.

Answer: Thank you for your professional suggestion! We have added the experimental details of the TGA and the Raman experiments in 3.3 Catalyst characterization part.

Thermogravimetric analysis (TGA) was used to characterize the carbon deposition of used catalysts. The sample (10 mg) was treated under air atmosphere with a flow rate of 30 mL/min-1 at room tempeature until the scales balanced. Then, the temperature increased from room temperature to 800 °C with a heating rate of 5 °C min-1. The data was recorded (the weight of sample) from room temperature to 800 °C.

Raman spectroscopy measurements were carried out on an objective (X50LWD) with a Filter of D1, a Hole of 200 µm, a Grating of 600 gr/mm and a Laser of 532.17 nm. The wavenumber values were scaned over the range 40−4000 cm-1for three times.

I also believe that the quality of the manuscript would be improved if CO adsorption tests were done as well. This would give the authors information about the Ni dispersion as a function of Y addition. This group seems to have all of the necessary equipment to perform such an experiment based on the CO2/N2 adsorption results already presented in the paper.

Answer: Thanks a lot for this suggestion that can be very interesting in order to have the percentage of metal exposed, we will add this technique in our next manuscript.

Reviewer 3 Report

Dry reforming of methane has been investigated comparing a Y-doped with a Y-free NiO-ZrOx catalyst. The work concentrates on changes of the catalyst system during reaction, especially carbon deposition.

In general, short reaction times of 8 hours (Fig. 4: 5 hours?) have been chosen so that validity of the obtained results, regarding in particular catalyst stability, is somewhat limited.

The following topics are also important since readability is affected:

l. 85: “… a high resistance against carbon deposition.”

Table 1:

- Please specify symbols and abbreviations.

- “calculated by” suggests that only calculations have been carried out and should be substituted by “determined by” (here and throughout the manuscript).

l. 127: “… consumes more H2 …”

l. 129: “… reduced from …”

Figure 1(B), x-axis: “Binding Energy” (see also Figure 5(A))

l. 150: “… one can note …”

Figure 3: An assignment of the curves is missing.

Table 3:

- “Crystallite sizes”

- Units are missing for the first columns.

- Footnote d: “Raman spectroscopy”

l. 183: “… was investigated at …”

Figure 4, caption: “… the H2/CO ratio …”

l. 222: Please specify the Ni particle sizes (unit is missing).

l. 227 - l. 231: Please discuss Fig. 6 more precisely. Is the formation of carbon nanotubes unambiguous? Is there really no carbon deposition on the Y-doped catalyst? Maybe it is not visible on the picture but nonetheless present.

l. 244: At which temperature did you start heating?

l. 273: “… equipped with …”

Author Response

Reviewer 3:

Dry reforming of methane has been investigated comparing a Y-doped with a Y-free NiO-ZrOx catalyst. The work concentrates on changes of the catalyst system during reaction, especially carbon deposition.

In general, short reaction times of 8 hours (Fig. 4: 5 hours?) have been chosen so that validity of the obtained results, regarding in particular catalyst stability, is somewhat limited.

Answer: Thank you very much for your kind suggestion! We had modified Figure 4 in the manuscript. For the stability, we couldn't do the experiment at night for safety reasons. So, we just run 8 hours. In this system, we focuss on the ability to restrain coke. We can find that yttrium can enhance the ability of restrain coke even reaction for 8 hours.

Figure 4 The conversion of CH4 (A) and CO2 (B), and the H2/CO ratio (C) of NiO-ZrOm-YOn and NiO-ZrOm catalysts at 700 oC with the mixed flow of CH4:CO2:Ar=10:10:80, GSHV=48,000 h-1.

The following topics are also important since readability is affected: 85: “… a high resistance against carbon deposition.” 127: “… consumes more H2 …” 129: “… reduced from …” 150: “… one can note …” 183: “… was investigated at …” 273: “… equipped with …”

Answer: Thank you very much for your kind suggestion! We had modified in related part in the manuscript.

Table 1:

- Please specify symbols and abbreviations.

- “calculated by” suggests that only calculations have been carried out and should be substituted by “determined by” (here and throughout the manuscript).

Answer: Thank you for pointing out this point! We had modified Table 1 and changed the word of “calculated by” throughout the manuscript. We also specified the symbols and abbreviations.

Table 1 The results of the BET experiment for NiO-ZrOm and NiO-ZrOm-YOn calcined catalysts, the specific surface area (SBET) was determined by BET method; the pore volume (VP) and the pore diameter (DP) determined by BJH method. H2 consumption of calcined catalysts determined by H2-TPR. The content of Zr4+ and Zr3+ on both catalysts after reduction determined by XPS. And the content of Ni on both catalysts determined by ICP.

Catalyst

DP,

nm

SBET, m2/g

VP, cm3/g

H2 consumption, mmol H2/g

Zr 3d (%)

Ni

(wt %)

Total

Theorya

α

β

Zr4+

Zr3+

NiO-ZrOm-YOn

2

79

0.1

0.37

0.39

0.10

0.27

16

84

10

NiO-ZrOm

2

113

0.2

0.62

0.47

0.13

0.49

10

90

12

a : Caculated by ICP and the comsuption of reduction of pure NiO

Figure 1(B), x-axis: “Binding Energy” (see also Figure 5(A))

Answer: Thanks for pointing out our error! We had modified Figures 1 and 5.

Figure 1 (A) the H2-TPR profiles of NiO-ZrOm-YOn and NiO-ZrOm calcined catalysts, and (B) the Zr 3d profiles from XPS measurements of NiO-ZrOm-YOn and NiO-ZrOm catalysts after reduction.

Figure 5 The C 1s (A) and Raman (B) profiles of used NiO-ZrOm and NiO-ZrOm-YOn catalysts.

Figure 3: An assignment of the curves is missing.

Answer: Thanks for your nice comments, we missed some clear description. We had modified Figure 3 in the manuscript in order to make it to be clearer.

Figure 3 The XRD profiles of (a) NiO-ZrOm and (c) NiO-ZrOm-YOn catalysts after reduction at 700 °C for 1 hour and after reaction at 700 °C for 8 hours, and (b) NiO-ZrOm and (d) NiO-ZrOm-YOn catalysts after reaction at 700 °C for 8 hours.

Table 3:

“Crystallite sizes”

Units are missing for the first columns.

Footnote d: “Raman spectroscopy”

Answer: Thanks for your nice comments, we had modified Table 3 in the manuscript.

Table 3 Crystallite sizes of ZrO2 and Ni0 on NiO-ZrOm-YOn and NiO-ZrOm catalysts after reduction at 700 °C for 1 hour and after reaction at 700 °C for 8 hours, determined by XRD. The particle size of nickel after reaction for 8 hours determined by TEM. The coke content after reaction for 8 hours determied by TG. The C-C content after reaction for 8 hours determined by XPS. The IG/ID after reaction for 8 hours determined by Raman spectroscopy.

Catalyst

ZrO2 (nm)

Ni0 (nm)

Particle

Coke

Content

(%)

C-C

content (%)

IG/ID

Reduction

Reaction

Reduction

Reaction

Size (nm)

NiO-ZrOm

7

7

12

24

15-20

3.7

42

1.7

NiO-ZrOm-YOn

7

7

16

10

10-15

1.0

38

-

Figure 4, caption: “… the H2/CO ratio …”

Answer: Thanks for your nice comments, we had modified the caption of Figure 4 in the manuscript.

Figure 4 The conversion of CH4 (A) and CO2 (B), and the H2/CO ratio (C) of NiO-ZrOm-YOn and NiO-ZrOm catalysts at 700 oC with the mixed flow of CH4:CO2:Ar=10:10:80, GSHV=48,000 h-1.

222: Please specify the Ni particle sizes (unit is missing).

Answer: Thanks for your nice comments, we had added the unit in the manuscript.

It can be noted that large Ni particle (over 20 nm) can be observed on NiO-ZrOm catalyst.

227 - l. 231: Please discuss Fig. 6 more precisely. Is the formation of carbon nanotubes unambiguous? Is there really no carbon deposition on the Y-doped catalyst? Maybe it is not visible on the picture but nonetheless present.

Answer: Thank you very much for your kind suggestion! We had modified the related part in the manuscript. These large nickel particles maybe favor for selective reactions that tend to form carbon deposition [1, 2], thereby resulting in formation of carbon deposition. Thus, a lot of carbon deposition can be observed on NiO-ZrOm catalyst in the form of carbon nano tube (Fig 6 (A)), and no carbon deposition is observed on NiO-ZrOm-YOn catalyst (Fig 6 (C)), which indicates that no or little carbon formed on NiO-ZrOm-YOn catalyst. These carbon nanotubes are graphitic carbon, which is in agreement with the results of Raman experiment. This phenomenon shows that yttrium can suppress the sintering of nickel particles, and also inhibits the formation carbon during the DRM reaction. We have modified the relevant part of the manuscript.

Figure 6 TEM images of used catalysts (A) NiO-ZrOm and (C) NiO-ZrOm-YOn and particle sizes of nieckel on catalysts (B) NiO-ZrOm and (D) NiO-ZrOm-YOn.

244: At which temperature did you start heating?

Answer: Thanks for your nice comments. The mixed liquor was heated from room temperature to 95 °C with constant stirring for 48 h. We had revised the relevant part of the manuscript.

Reference:

[1] Sutthiumporn K., Kawi S., Promotional effect of alkaline earth over Ni–La2O3 catalyst for CO2 reforming of CH4: role of surface oxygen species on H2 production and carbon suppression, Int. J. Hydrogen. Energy, (2011) 36, 14435-14446.

[2] Świrk K., Gálvez M.E., Motak M., Grzybek T., Rønning M., Costa P. Da, Syngas production from dry methane reforming over yttrium-promoted nickel-KIT-6 catalysts, Int. J. Hydrogen. Energy, 44 (2019) 274-286

Round 2

Reviewer 1 Report

Overall I am satisfied with the reviewers changes. They have responded well to the feedback and the paper is improved signicantly. 

Reviewer 3 Report

After revision, quality of the manuscript improved remarkably.